# Two Novel *Microbacterium* Species Isolated from *Citrullus colocynthis* L. (Cucurbitaceae), a Medicinal Plant from Arid Environments

**DOI:** 10.3390/microorganisms13112583

**Published:** 2025-11-12

**Authors:** Khadija Ait Si Mhand, Salma Mouhib, Nabil Radouane, Khaoula Errafii, Issam Meftah Kadmiri, Derly Madeleiny Andrade-Molina, Juan Carlos Fernández-Cadena, Mohamed Hijri

**Affiliations:** 1African Genome Center, University Mohammed VI Polytechnic (UM6P), Lot 660, Hay Moulay Rachid, Ben Guerir 43150, Morocco; khadija.aitsimhand@um6p.ma (K.A.S.M.); salma.mouhib@um6p.ma (S.M.); nabil.radouane-ext@um6p.ma (N.R.); khaoula.errafii@um6p.ma (K.E.); 2Plant and Microbial Biotechnology Center, Moroccan Foundation for Advanced Science, Innovation and Research (MAScIR), University Mohammed VI Polytechnic (UM6P), Lot 660, Hay Moulay Rachid, Ben Guerir 43150, Morocco; issam.kadmiri@um6p.ma; 3OMICS Sciences Laboratory, Faculty of Health Science, Universidad Espíritu Santo, Samborondón 092301, Ecuador; dmandrademolina@uees.edu.ec (D.M.A.-M.); jfernandezcadena@bwh.harvard.edu (J.C.F.-C.); 4Harvard Medical School, Brigham and Women’s Hospital, Boston, MA 02115, USA; 5Institut de Recherche en Biologie Végétale (IRBV), Département de Sciences Biologiques, Université de Montréal, 4101 Rue Sherbrooke Est, Montréal, QC H1X 2B2, Canada

**Keywords:** arid environment, endophytic bacteria, Cucurbitaceae, *Microbacterium*, plant–microbe interaction, whole-genome sequencing

## Abstract

Plants inhabiting arid regions often harbor microbial communities that contribute to their resilience under extreme conditions. Yet, the genomic diversity and functional potential of bacterial endophytes associated with desert-adapted plants, particularly in Africa, remain largely unexplored. In this study, we investigated *Microbacterium* endophytes from the xerophytic cucurbit *Citrullus colocynthis* L. (Cucurbitaceae), collected in a semi-arid environment in central Morocco. Using culture-based isolation, phenotypic characterization, and whole-genome sequencing, we analyzed three representative isolates from leaf and root tissues. Genome-based taxonomy combined with polyphasic analyses identified two novel species, *Microbacterium xerophyticum* sp. nov. and *Microbacterium umsixpiens* sp. nov., with genome sizes of approximately 4.0 Mb and 3.9 Mb, respectively. Functional annotation revealed traits consistent with endophytism in water-limited ecosystems, including oxidative and osmotic stress responses, metal homeostasis, and high-affinity phosphate uptake. Differences in siderophore acquisition and nitrogen metabolism suggest niche partitioning between the two species. These findings document two novel bacterial species from a medicinal plant native to arid ecosystems, broaden the known diversity of plant-associated *Microbacterium*, and provide region-specific genomic references with adaptive traits relevant to host resilience under arid conditions.

## 1. Introduction

Drylands, comprising arid and semi-arid ecosystems, cover more than 40% of the Earth’s terrestrial surface and sustain nearly two billion people worldwide [1]. These regions are shaped by extreme environmental pressures, including high temperatures, irregular and scarce rainfall, recurrent droughts, intense solar and ultraviolet radiation, and poor soil fertility [2]. Such stressors not only constrain biodiversity but also make drylands among the most fragile ecosystems on the planet. In Morocco, desertification is intensifying due to the combined effects of climate change and anthropogenic pressures, particularly overgrazing and unsustainable land use [3,4]. Rising temperatures and declining precipitation patterns are accelerating land degradation and threatening agriculture, biodiversity, and local livelihoods [5,6]. Beyond regional consequences, the degradation of these ecosystems has global implications through altered carbon and nutrient cycles, reduced agricultural productivity, and increased vulnerability to climate change [4,7].

Despite these challenges, Morocco’s arid and semi-arid landscapes host resilient xerophytic vegetation adapted to water scarcity and nutrient-poor soils [8]. The Marrakech–Safi region, for instance, supports a mosaic of drought-tolerant plants that thrive in microhabitats where soil moisture, organic matter, and microbial biomass are relatively enriched [9,10,11]. These microhabitats not only provide ecological niches for plants but also support diverse microbial communities that may contribute to plant adaptation under stress. Increasing evidence suggests that plant–microbe interactions are central to survival in drylands, with microbial endophytes acting as hidden allies that improve plant resilience to abiotic and biotic stresses [12].

Among these plants, *Citrullus colocynthis* (L.) Schrader (Cucurbitaceae), also known as bitter apple, colocynth, or desert gourd, is widely distributed across the Sahara, Middle East, and Indian subcontinent, and is particularly abundant in Morocco’s semi-arid ecosystems [13]. This perennial creeping vine has long been recognized for its medicinal properties; extracts from its roots, fruits, and seeds have been used in traditional medicine for their purgative, anti-inflammatory, and antimicrobial effects [14]. Phytochemical studies have revealed a rich arsenal of bioactive compounds, including cucurbitacins, alkaloids, flavonoids, and glycosides, which contribute to its medicinal and ecological resilience [15]. Beyond chemical defenses, the ability of *C. colocynthis* to survive under extreme conditions suggests potential reliance on microbial symbionts that enhance stress tolerance and nutrient acquisition. Endophytic microbes colonizing *C. colocynthis* may therefore play crucial roles in shaping its ecological success and adaptive strategies.

The family *Microbacteriaceae* (phylum *Actinobacteria*) includes small, Gram-positive, non-spore-forming bacteria with high G + C content. Members of the genus *Microbacterium* have been isolated from diverse habitats such as soils, freshwater, marine sediments, industrial effluents, and, importantly, plant tissues [16,17]. Several species are known as plant endophytes, where they contribute to host fitness by promoting growth, improving nutrient acquisition, and enhancing stress tolerance [18]. *Microbacterium* spp. are versatile metabolically and are reported to degrade xenobiotics, tolerate heavy metals, and survive in nutrient-limited conditions, traits that likely support their persistence in extreme environments [19].

The contribution of *Microbacterium* to plant growth promotion (PGP) has been demonstrated in several studies [20,21,22,23,24]. These bacteria employ both direct and indirect mechanisms. Direct mechanisms include nitrogen fixation, phosphate solubilization, production of phytohormones (auxins, cytokinins), siderophore production, and ACC deaminase activity, all of which enhance nutrient availability, stimulate root development, and mitigate ethylene stress [25]. Indirect mechanisms involve the induction of systemic resistance, antioxidant responses, and tolerance to abiotic stresses such as drought, salinity, and heavy metal contamination [26]. For example, *M. imperiale* and *M. paraoxydans* have been shown to induce systemic responses in host plants, thereby increasing resilience to heavy metal stress [27,28]. Beyond growth promotion, *Microbacterium* spp. also contribute to phytoremediation by supporting plants in contaminated soils, enabling them to tolerate and remediate pollutants [18,29]. Together, these features highlight the ecological and biotechnological significance of the genus in sustainable agriculture and environmental restoration, particularly in fragile ecosystems such as drylands.

Despite their importance, genomic insights into plant-associated *Microbacterium* remain limited, especially in Africa. Most studies involving *C. colocynthis* and other local xerophytes rely on cultivation-dependent methods or 16S rRNA gene surveys, which provide limited taxonomic and functional resolution [30,31]. Genome-resolved studies of *Microbacterium* from plant origins are available outside Africa, for example, isolates have been reported from grapevine xylem, rice seeds, and potato [30,32,33]. By contrast, in North Africa, genome-sequenced *Microbacterium* strains have been described only from industrial effluents, such as *M. metallidurans* TL13 (from Tunisian leather effluent) [34], *Microbacterium* sp. Nx66 (from Algerian petrochemical effluent) [35], and *M. algeriense* (from Algerian oil-production water) [36]. This imbalance highlights a striking knowledge gap: there is no genomic reference for *Microbacterium* directly isolated from plant hosts in African arid or semi-arid ecosystems.

Establishing genome-defined species from *C. colocynthis* in Morocco’s semi-arid environments provides valuable region-specific references for both taxonomic and ecological studies. Such references not only expand the recognized diversity of plant-associated *Microbacterium* but also enable deeper understanding of their adaptive traits and potential roles in plant resilience under extreme conditions.

In this study, we describe the genome-based characterization of two novel *Microbacterium* species isolated from root and foliar tissues of *C. colocynthis* collected in central Morocco. Using culture-based isolation, phenotypic characterization, and whole-genome sequencing, we determined their taxonomic status through average nucleotide identity (ANI), digital DNA–DNA hybridization (dDDH), and phylogenomic inference. Genomic features, including genome size, G + C content, and coding potential, were also analyzed. Functional annotation was carried out to identify traits associated with plant colonization and adaptation to arid ecosystems. Our results support the recognition of *Microbacterium xerophyticum* sp. nov. and *Microbacterium umsixpiens* sp. nov., expanding the diversity of plant-associated actinobacteria and providing genomic insights into their potential ecological functions in arid environments.

## 2. Materials and Methods

### 2.1. Sampling and Isolation

Three *Microbacterium* strains were isolated from the roots and leaf tissues of *Citrullus colocynthis* collected at the flowering stage along a decommissioned wastewater canal near Green City, Benguerir, Morocco (32°11′48.6″ N, 7°56′30.0″ W) on 23 July 2023. Plants were transported on ice and processed immediately. Root and foliar tissues were surface-sterilized, with sterility confirmed by imprint tests and final rinse controls. Sterilized tissues were cut into 5 × 5 mm fragments and plated on Tryptone Soya Agar (TSA) medium (per liter: pancreatic digest of casein 17 g, papaic digest of soybean meal 3 g, NaCl 5 g, K_2_HPO_4_ 2.5 g, glucose 2.5 g, and agar 15 g) (Oxoid, Thermo-Fisher, Temara, Morocco), both at full-strength and 1/10 dilution, with one quadrant supplemented with sugar-free minimal medium. Plates were incubated at 28 °C for up to 4 weeks, after which distinct colonies were purified and cryopreserved in TSB containing 25% (*v*/*v*) glycerol at −80 °C (Figure 1).

### 2.2. DNA Extraction, 16S rRNA Gene Sequencing and Whole-Genome Sequencing

Genomic DNA from each isolate was extracted from fresh bacterial suspensions [37]. Isolates were initially identified by PCR amplification and Sanger sequencing of the 16S rRNA gene [29,37]. For whole-genome sequencing, DNA from the three isolates was quantified, and libraries were prepared using a Nextera XT kit (Illumina, MegaFlex, Casablanca, Morocco) following the manufacturer’s protocol (200 ng input per sample; unique dual indexes). Libraries were loaded and sequenced individually, achieving a cluster density of 202 K/mm^2^. Sequencing was performed in a 2 × 151 bp paired-end format. The resulting data were demultiplexed with bcl2fastq (v2.17.1.14; Illumina) to ensure accurate sample identification and assignment for downstream analysis. This workflow enabled efficient, high-precision bacterial genome sequencing.

### 2.3. Bioinformatics Workflow

Figure 2 summarizes the bioinformatics pipeline, from raw sequence data to species delineation. The subsections below detail the exact tools, parameters, and thresholds applied at each step.

### 2.4. Raw Data Preprocessing and Quality Assessment

Raw sequencing reads were preprocessed using BBDuk74 to remove low-quality sequences (Q ≤ 20) and short reads (minimum length = 100 bp). To assess potential contamination from non-target bacterial species, the filtered reads were analyzed with Kaiju for taxonomic classification [38].

### 2.5. Genome Assembly, Completeness, and Optimization

*De novo* assembly of each sample was performed using MaSuRCA v4.1.4 [39]. Assemblies were evaluated with QUAST v5.3.0 [40], and the versions with the fewest contigs and highest N50 values were selected for further analysis. Genome completeness was estimated using CheckM v1.2.0 [41]. In cases where Kaiju v1.10.1 detected contamination and assembly metrics were suboptimal, metaSPAdes v4.2.0 [42] was applied. Prior to assembly, MaxBin v2.2.7 [43] was used to separate potential contaminant bins, and DAS Tool v1.1.7 [44] was employed to refine bins based on sequence similarity. Final assemblies were again assessed with QUAST v5.3.0, and the most complete, least fragmented genome was selected for downstream analysis.

### 2.6. Taxonomic Assignment

For post-assembly taxonomic classification, isolates were analyzed using the Type (Strain) Genome Server (TYGS) [45] and PubMLST; https://pubmlst.org accessed on 3 November 2025 [46]. TYGS compared whole-genome and 16S rRNA sequences against type strains to identify closely related taxa and assess the novelty of the isolates. PubMLST indexed up to 53 ribosomal protein subunit genes (*rps* genes) for multilocus sequence typing (MLST), reporting support percentages at a 95% confidence threshold relative to the closest reference strains.

### 2.7. Phylogenetic Characterization, Species Inference, and Endophytic Traits

To evaluate the novelty of the isolates, species boundaries were inferred using ANI (average nucleotide identity) with a 96% threshold [47] and digital DNA–DNA hybridization (dDDH) with a 70% cutoff via the Genome-to-Genome Distance Calculator [48]. Whole-genome-based phylogenetic relationships were inferred using the Type (Strain) Genome Server (TYGS; https://tygs.dsmz.de, accessed on 5 November 2025). Draft assemblies of the novel *Microbacterium* strains, the type strains of the closest *Microbacterium* species identified in ANI/dDDH analyses, and the outgroup *Leifsonia xyli* DSM 46,306 were analyzed with the Genome BLAST Distance Phylogeny (GBDP) method under default parameters. Phylogenetic inference was performed with FastME on the GBDP distance matrix, with branch lengths representing intergenomic distances and node support values estimated from 100 pseudo-bootstrap replicates. The resulting tree was visualized and manually rerooted on the outgroup in iTOL.

Endophytic traits were investigated by annotating genomes with Prokka v1.14.6 [49], followed by KEGG Orthology assignment using KofamKOALA [50]. A curated list of genes linked to endophytic functions was applied to identify and classify candidate genes [51,52].

### 2.8. Draft Genome Visualization and Comparative Analysis

Draft genomes were visualized with Proksee; https://proksee.ca accessed on 3 November 2025, highlighting GC content, GC skew, mobile elements, and resistance genes based on MobileOG-db 1.1.3 [53] and CARD [54]. Comparative genomic analyses were performed using the RAST; https://rast.nmpdr.org (accessed on 3 November 2025) subsystems database [55] to identify core metabolic pathways and functional genes. Normalized data were used to construct heatmaps in RStudio (R 4.3.2) with the pheatmap package (version 1.0.12) [56].

### 2.9. Phenotypic Charcterization of Novel Species

Phenotypic profiling was conducted with the Biolog GEN III system (Biolog Inc., Hayward, CA, USA), following the manufacturer’s guidelines. This system assessed carbon source utilization and chemical sensitivity. Absorbance at 590 nm was recorded with a Biolog MicroStation™ at 24 h intervals for up to seven days. The resulting profiles were compared to the Biolog GEN III database for species-level characterization [57].

### 2.10. Genome Alignment Visualization

Pairwise whole-genome alignments were generated with Mauve (progressiveMauve module; default scoring 2.4.0) using default scoring parameters (including seed weight and minimum LCB size). Draft assemblies (AGC47, AGC85, AGC62) in FASTA format were aligned as AGC47 vs. AGC85 and AGC62 vs. AGC85. Locally collinear block weight (LCB w) were identified and visualized with the Mauve viewer, where inverted segments appear in reverse orientation. The alignment plots were exported directly from Mauve as PNG images [58,59].

## 3. Results

### 3.1. Isolation and Characterization of Bacaterial Endophytes

Three *Microbacterium* isolates were obtained: AGC47 (leaf; Cc-F), AGC85 (root; Cc-R), and AGC62 (root; Cc-R). Cc-F = *Citrillus colocynthis,* foliar tissue; Cc-R = *Citrillus colocynthis,* roots tissue. AGC47 was isolated from a distinct *C. colocynthis* plant sample, whereas AGC62 and AGC85 originated from the same plant (root and leaf, respectively). Comparative genomic analyses supported the recognition of two novel species: *M. xerophyticum* sp. nov. (AGC47, AGC85) and *M. umsixpiens* sp. nov. (AGC-62).

### 3.2. Genome Features of Novel Microbacterium Strains

*M. umsixpiens* AGC62 had a genome size of 3.91 Mb, with a genome completeness of 99.17% and contamination of 1.01% (Appendix A). MLST analysis indicated a 66% relatedness to *M. liquefaciens* (GCF_024362265.1). ANI (86.5%) and dDDH (44.4%) values confirmed its novelty at the species level, and no alignment was detected with TYGS type strains (Table 1).

*M. xerophyticum* sp. nov. (AGC47, AGC85) showed a genome completeness of 99.19% and 97.47%, with contamination levels of 1.06% and 0.61%, respectively (Appendix A). The two genomes share 99.99% ANI. Both isolates clustered with *M. profundi* (GCF_000763375.1) in MLST analysis but fell below species-level thresholds (ANI: 89.1%; dDDH: 33%) (Table 1).

A detailed overview of assembly approaches, genome statistics, quality assessments, coding sequences (CDSs), tRNA counts, 16S rRNA gene recovery, and SRA accession numbers is provided in Appendix A.

### 3.3. Gene Content

Across the three genomes, we identified a conserved set of stress-response and nutrient-acquisition genes (Figure 3). Oxidative stress management modules (e.g., thioredoxin/thioredoxin-reductase), metal homeostasis and efflux systems, osmoprotection mechanisms, and phosphate scavenging (e.g., high-affinity phosphate uptake, polyphosphate metabolism) were consistently present. Siderophore biosynthesis and uptake loci were irregularly distributed. None of the genomes encoded a complete denitrification pathway; nitrogen metabolism was restricted to assimilatory nitrate/nitrite reduction. These features collectively suggest adaptation to oxidative and osmotic stress and nutrient limitation, particularly phosphorus scarcity, typical of semi-arid ecosystems.

### 3.4. Taxonomic Assignment

Taxonomic classification using MLST (PubMLST; https://pubmlst.org accessed on 3 November 2025) analysis confirmed that the isolates were distinct from described *Microbacterium* species. Strains AGC47 (leaf) and AGC85 (root) grouped together as *M. xerophyticum* sp. nov., sharing 100% MLST similarity and showing *M. profundi* as their closest relative (ANI = 89.1%; dDDH = 33%), values below the species thresholds (ANI ≥ 95%; dDDH ≥ 70%). Strain AGC62 (root) was classified as *M. umsixpiens* sp. nov., showing 66% MLST similarity, 86.5% ANI, and 44.4% dDDH with *M. liquefaciens*. TYGS analysis did not return a match to any type strain for the three isolates. These results support the delineation of two novel *Microbacterium* species associated with *C. colocynthis* (Table 1).

### 3.5. Nomenclature and Biochemical Profiling of Novel Bacterial Species

#### 3.5.1. *Microbacterium umsixpiens* sp. nov.

*Microbacterium umsixpiens AGC62* was isolated from the roots of *C. colocynthis.* Growth occurred at 28–30 °C on TSA medium after 24 h. Colonies were egg-yellow, round, convex, and viscoid, with a diameter of approximately 2 mm. Genome-inferred chemotaxonomic profiles (peptidoglycan, menaquinone, polar lipid, cell-wall sugar, fatty-acid pathways) are summarized in Appendix A.

**Etymology**: M.L. neut. adj. *umsixpiens*; derived from “UM6P,” the institution where the strain was isolated.

Type strain: AGC62^T (=CCMM B1339T).Accession numbers: 16S rRNA gene, PV706301; whole genome, SRR29855759.BioProject: PRJNA1133887; Biosample: SAMN42389299.DNA G + C content: 68.54%; Genome size: 3.91 Mb.ANI/closest relative: 86.5%. dDDH: 44.4%.

Phenotypic profiling revealed 93.8% similarity to *M. maritypicum*. The strain displayed broad metabolic versatility, utilizing diverse carbohydrates (dextrin, D-maltose, D-trehalose, D-cellobiose, sucrose, α-D-lactose), organic acids (D-gluconic acid, D-saccharic acid), and amino acids (L-alanine, L-aspartic acid, L-glutamic acid, L-histidine). It grew at pH 5–6, tolerated NaCl up to 8%, and resisted troleandomycin, rifamycin, nalidixic acid, and aztreonam. The draft genome and metabolic profile are presented in Figure 4.

Comparative phenotypic profiles of *Microbacterium umsixpiens* AGC62 and its closest related-type strain, *Microbacterium liquefaciens* DSM 20,638 ((Collins et al. [60]) Takeuchi and Hatano [61], Strain DSM 206338), are summarized in Appendix A.

#### 3.5.2. *Microbacterium xerophyticum* sp. nov.

*Microbacterium xerophyticum* AGC47 (leaf) and AGC85 (root) were isolated from *C. colocynthis*. Growth occurred at 28–30 °C on TSA medium after 24 h. Colonies were neon yellow, round, pulvinate, and smooth, with a diameter of approximately 2 mm. Genome-inferred chemotaxonomic profiles (peptidoglycan, menaquinone, polar lipid, cell-wall sugar, fatty-acid pathways) are summarized in Appendix A.

**Etymology**: M.L. neut. adj. *xerophyticum*; from Greek “xeros” (dry) + “phyton” (plant), reflecting its adaptation to arid environments as an endophyte.

Specie name: *Microbacterium xerophyticum* sp. nov.Type strain: AGC85^T (=CCMM B1344T).Accession numbers: 16S rRNA gene, PV706316; whole genome, SRR29855758.BioProject: PRJNA1133887; Biosample: SAMN42389300.DNA G + C content, 67.23%; Genome size, 4.06 Mb.ANI/closest relative: 89.1%. dDDH: 33%.

The biochemical profile of AGC85 revealed metabolic specialization, with the ability to metabolize D-fucose, L-fucose, L-rhamnose, sodium lactate, and several organic acids (pectin, D-galacturonic acid, L-galactonic acid lactone, D-glucuronic acid, glucuronamide, quinic acid). Growth was supported at pH 6 and up to 4% NaCl. The strain was resistant to nalidixic acid, aztreonam, lithium chloride, potassium tellurite, and sodium bromate. The draft genome and biochemical profile are shown in Figure 5.

Comparative phenotypic profiles comparing *Microbacterium xerophyticum* AGC85 and its closest related-type strain, *Microbacterium profundi* DSM 22,239 Wu et al. [62]), are summarized in Appendix A.

### 3.6. Phylogenetic Analysis

A species-level phylogeny was inferred using *Leifsonia xyli* DSM 46,306 as the outgroup. Branch support values are indicated at nodes, and scale bars represent substitutions per site (Figure 6). The tree resolved two distinct clades: AGC47 and AGC85 formed a monophyletic branch consistent with *M. xerophyticum* sp. nov., closely related to *M. profundi*; AGC62 formed an independent branch corresponding to *M. umsixpiens* sp. nov., closest to *M. aquaticae*. These relationships corroborate ANI and dDDH results, confirming species-level distinctiveness.

### 3.7. Genome Alignement of M. umsixpiens and M. xerophyticum (AGC62 and AGC85) with Either Closest Bacterial Relatives Using Mauve

Whole-genome alignment using Mauve v 2.4.0 revealed conserved chromosomal synteny in both novel species, AGC62 and AGC85, when compared with their closest relatives, *Microbacterium liquefaciens DSM* and *Microbacterium profundi* strain Shh49, respectively (Figure 7). After contig reordering according to reference position, *M. umsixpiens* AGC62 exhibited extensive collinearity with *M. liquefaciens DSM*, illustrated by and LCB weight of 266 (Figure 7A), with the corresponding maximum LCB w. Similarly, *M. xerophyticum* AGC85 displayed comparable conservation with *M. profundi* strain Shh49 (91 LCB w) (Figure 7B), and its associated maximum LCB w.

These alignment patterns indicate extensive genome collinearity with limited rearrangements, mainly short inversions and transpositions, concentrated near terminal regions, and only a few internal breakpoints. No major insertions or deletions were observed within the core genomic backbone, consistent with their classification as novel species based on ANI and TYGS analyses. The observed patterns are congruent with ANI values and TYGS phylogenetic placement of both isolates, supporting their designation as novel *Microbacterium* species exhibiting slight structural divergence.

### 3.8. Inter-Species Genome Restructuring Between M. xerophyticum and M. umsixpiens

Comparative analysis of *M. xerophyticum* (used as the reference genome) and *M. umsixpiens* revealed extensive genome reorganization (Figure 8). Mauve resolved the chromosomes into 454 small locally collinear blocks (LCBs), with the corresponding maximum LCB w. Frequent inversions and translocations were distributed throughout the chromosome, with only a few long collinear segments remaining. The dense crossings and fragmented synteny indicate genome-wide reshuffling of gene order. These patterns are consistent with genome-based species delineation (ANI, TYGS) and further support the classification of *M. umsixpiens* as a distinct species from *M. xerophyticum*.

## 4. Discussion

We report the isolation and genome-based characterization of three endophytic *Microbacterium* strains from *Citrullus colocynthis* collected in a semi-arid region of Morocco, resolving them into two novel species: *M. xerophyticum* sp. nov. (leaf and root isolates) and *M. umsixpiens* sp. nov. (root isolate). Their genome sizes (4.06/3.91 Mb), GC contents (67.23/68.54%), and coding sequence counts (3932/3996 CDS) are consistent with values reported for *Microbacterium* and other high-GC actinobacteria [63].

Whole-genome alignments reveal contrasting synteny patterns at two scales. Between *M. umsixpiens* AGC62 and *M. xerophyticum* AGC85, macro-synteny is highly fragmented with numerous inversions and transpositions, indicating broad structural divergence. On the other hand, the two *M. xerophyticum* strains (AGC47 vs. AGC85) are predominantly co-linear, with only a few localized inversions adjacent to contig boundaries in the draft assemblies. These visual patterns are consistent with our species delineation.

The three genomes shared a conserved repertoire of stress-response and nutrient-acquisition genes, including modules for oxidative stress defense, metal homeostasis, osmoprotection, and phosphate scavenging (high-affinity uptake and polyphosphate metabolism). Siderophore acquisition loci were detected in *M. xerophyticum*. Such traits are characteristic of plant-associated *Microbacterium*. For example, *M. testaceum* strains from rice carry phosphate-acquisition and siderophore genes and enhance host growth under stress [16]. In potato leaves, endophytic *Microbacterium* (e.g., StLB037) contribute to biocontrol via redox regulation and competition [64]. Likewise, grapevine endophyte *Microbacterium* che218 exhibits genomic features that support host metabolism in planta [65]. Comparative studies across 70 *Microbacterium* genomes further highlight conserved functions in phosphate use, siderophore production, secondary metabolite biosynthesis, and stress tolerance [63]. Our isolates therefore align well with the established functional profile of plant-associated *Microbacterium*.

A key difference between the two species concerns nitrogen metabolism. Genes for nitrate/nitrite assimilation were present in *M. xerophyticum* (AGC47, AGC85) but absent in *M. umsixpiens* (AGC62). This suggests potential niche partitioning: *M. xerophyticum* may utilize inorganic nitrogen inside plant tissues, whereas *M. umsixpiens* relies more on organic nitrogen, consistent with its expanded amino acid catabolism. Experimental validation, e.g., nitrate-reductase assays or growth on nitrate/nitrite as sole N sources, would help clarify these functional distinctions.

The occurrence of *M. xerophyticum* in both leaf and root tissues demonstrates its compartmental flexibility in *C. colocynthis*. Its genome indicates a core endophytic toolkit without strong compartment-specific specialization.

Biochemical profiles reinforced the ecological differentiation between the two species. *M. xerophyticum* (AGC85) showed selective metabolism of deoxy sugars (D-/L-fucose, L-rhamnose), pectin and uronic acids, quinic acid, and sodium lactate, substrates linked to plant cell wall turnover and phenolic metabolism [66,67,68], Its growth under moderate salinity (4% NaCl) and pH 6, and tolerance to lithium chloride, potassium tellurite, sodium bromate, and antibiotics, suggests adaptation to plant-derived polymers and oxidative/metal stresses. In contrast, *M. umsixpiens* (AGC62) exhibited broader metabolic versatility, utilizing diverse carbohydrates (dextrin, maltose, trehalose, cellobiose, sucrose, lactose), organic acids (gluconate, saccharate), and amino acids. Its higher NaCl tolerance (up to 8%) indicates greater osmoadaptation, potentially favoring persistence deeper in the root or rhizosphere.

Overall, these results suggest two complementary endophytic strategies in *C. colocynthis*: (i) *M. xerophyticum*, a host-metabolite specialist that exploits plant-derived sugars and acids in both root and leaf tissues, and (ii) *M. umsixpiens*, a generalist with broad substrate utilization and enhanced osmotic tolerance. Both species harbor stress-response and nutrient-acquisition traits that support persistence in water-limited, nutrient-poor tissues, while also contributing to host stress mitigation through redox balance and osmotic regulation.

## 5. Conclusions

This study identifies two novel endophytic *Microbacterium* species, *M. xerophyticum* sp. nov. (leaf and root) and *M. umsixpiens* sp. nov. (root), from the xerophytic plant *Citrullus colocynthis* in arid Morocco. Genome-based taxonomy (ANI, dDDH, phylogenomics) establishes them as distinct lineages, while functional profiles highlight capacities central to endophytism in drylands: oxidative and osmotic stress responses, metal homeostasis, phosphate uptake and polyphosphate metabolism, and siderophore acquisition. Differences in nitrogen metabolism and carbon use suggest ecological partitioning between a plant-metabolite specialist (*M. xerophyticum*) and a stress-tolerant generalist (*M. umsixpiens*).

By providing region-specific genomic references from a North African semi-arid ecosystem, this work strengthens microbial taxonomy in underexplored regions and advances understanding of *Microbacterium* adaptation to arid plants. The results also offer practical leads for applications: assays of phosphate solubilization, siderophore activity, nitrate/nitrite utilization, and salt/oxidative tolerance, followed by host inoculation tests, could evaluate these strains as bioregion-appropriate bioinoculants. By integrating taxonomy with functional genomics in a local context, this study contributes to microbial knowledge sovereignty and highlights the potential of indigenous endophytes to enhance crop resilience in arid agroecosystems. 

## Figures and Tables

**Figure 1 microorganisms-13-02583-f001:**
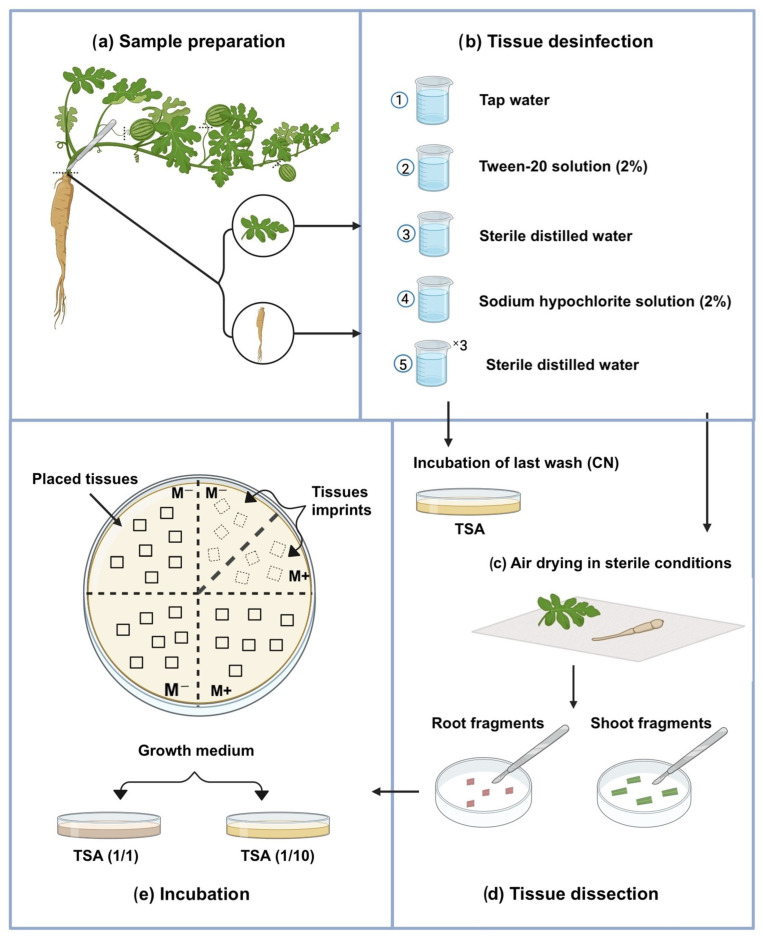
Workflow for the isolation process of endophytic bacteria. (**a**) Sample preparation. (**b**,**c**) tissue disinfection and drying. (**d**) tissue dissection. (**e**) growth of endophytic bacteria. CN: control negative; M+: TSA enriched with M medium; M−: TSA without M medium enrichment.

**Figure 2 microorganisms-13-02583-f002:**
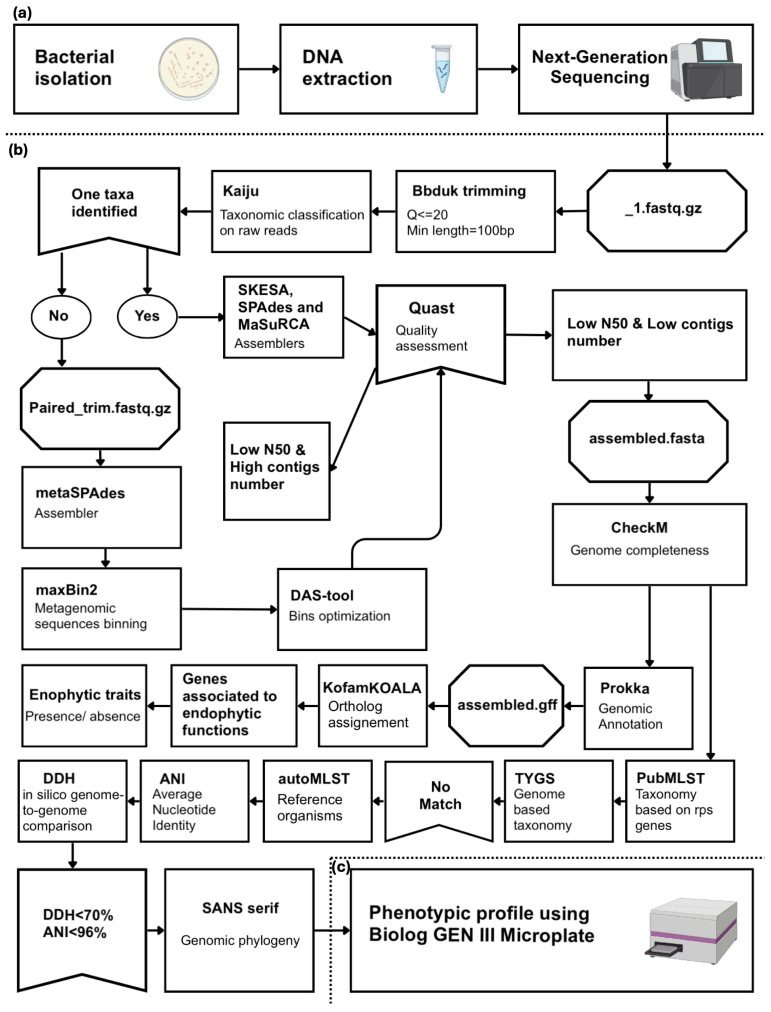
Experimental workflow and bioinformatic pipelines for identifying new species. (**a**) Pre-bioinformatic steps. (**b**) Bioinformatics workflow. (**c**) Phenotypic profiling.

**Figure 3 microorganisms-13-02583-f003:**
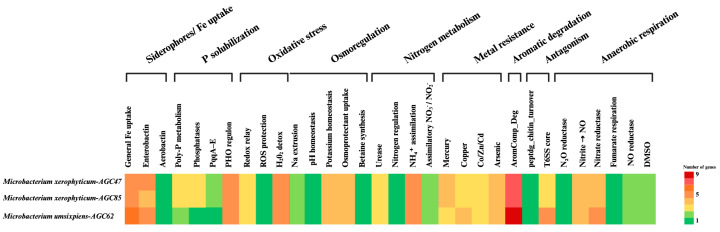
Heatmap shows the distribution of functional gene categories involved in plant growth promotion, stress response, and metabolism, based on gene count.

**Figure 4 microorganisms-13-02583-f004:**
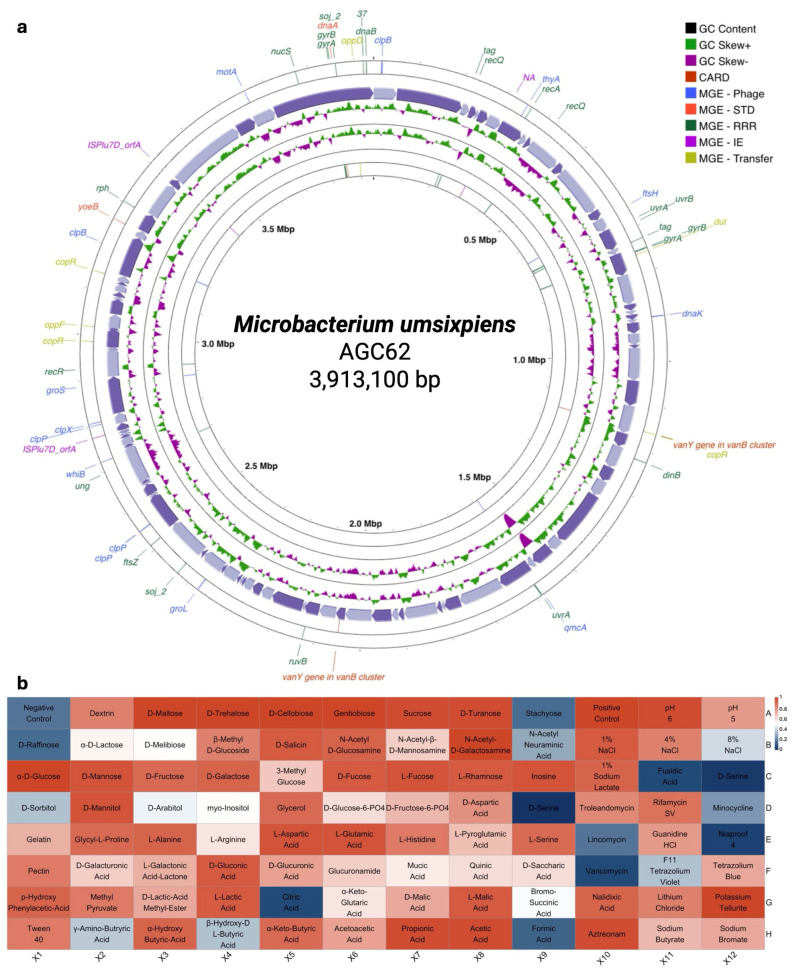
Genome map and biochemical profile of *Microbacterium umsixpiens* sp. nov. (**a**) The genome map, the outermost circles indicate genome annotations for mobile genetic elements (MGEs), categorized into Phage, Stability Transfer Defense (STD), Replication Recombination Repair (RRR), Integration Excision (IE), and Transfer. Purple arrows represent the contigs assembled in each sample, with the GC content and GC skew (±) displayed. The innermost circle highlights antimicrobial resistance annotations using the CARD database; https://card.mcmaster.ca accessed on 3 November 2025. (**b**) The biochemical profile, assessed using a GEN III plate type and analyzed using the Biolog MicroStation v4.20.05, *Microbacterium umsixpiens* sp. nov. strain AGC62, isolated from the root compartment in *Citrullus colocynthis*.

**Figure 5 microorganisms-13-02583-f005:**
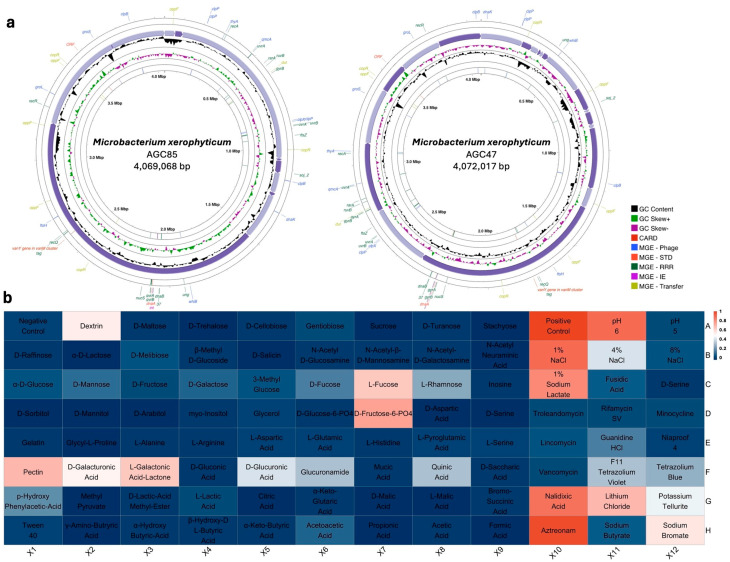
Genome map and biochemical profile of *Microbacterium xerophyticum* sp. nov. (**a**), The genome map, the outermost circles indicate genome annotations for mobile genetic elements (MGEs), categorized into Phage, Stability Transfer Defense (STD), Replication Recombination Repair (RRR), Integration Excision (IE), and Transfer. Purple arrows represent the contigs assembled in each sample, with GC content and GC skew (±) displayed. The innermost circle highlights antimicrobial resistance annotations using the CARD database. (**b**), The biochemical profile, assessed using a GEN III plate type and analyzed using the Biolog MicroStation.

**Figure 6 microorganisms-13-02583-f006:**
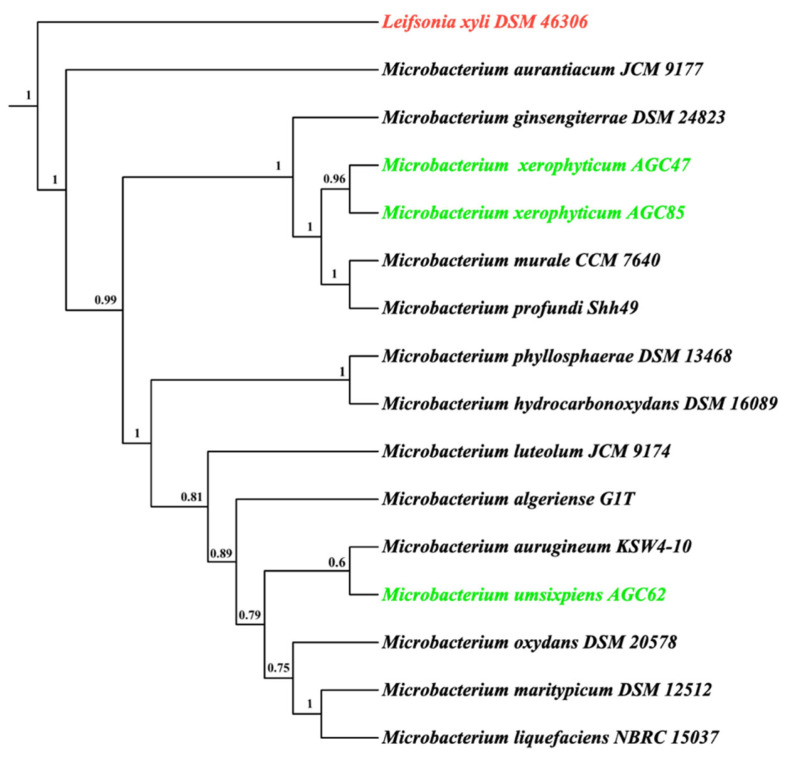
Phylogenomic tree of *Microbacterium* strains from *Citrullus colocynthis* showing the placement of novel isolates *Microbacterium xerophyticum* AGC47, AGC85, and *Microbacterium umsixpiens* AGC62. The outgroup *Leifsonia xyli* DSM 46306 is shown in red, the novel isolates are shown in green.

**Figure 7 microorganisms-13-02583-f007:**
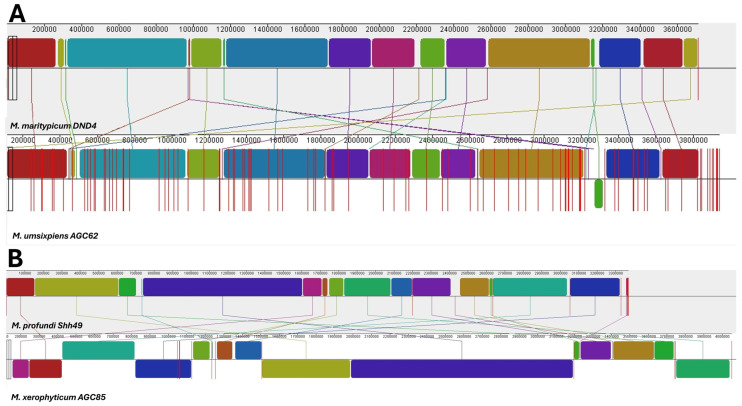
Whole-genome alignments of novel *Microbacterium* species with their closest relatives. (**A**) *M. umsixpiens* AGC62 vs. *M. liquefaciens DSM* (266 LCB w). (**B**) *M. xerophyticum* AGC85 vs. *M. profundi* Shh49 (91 LCB w). Contigs were ordered by reference position using Mauve.

**Figure 8 microorganisms-13-02583-f008:**
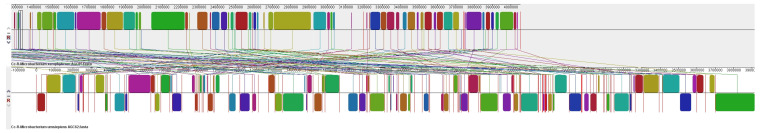
Inter-species genome restructuring between *M. xerophyticum* AGC85 and *M. umsixpiens* AGC62. Mauve alignment reveals extensive genome reorganization (454 LCB w), with contigs ordered according to the reference genome.

**Table 1 microorganisms-13-02583-t001:** Taxonomic classification of *Microbacterium* isolates from *Citrullus colocynthis*.

Plant Origin	Bacterial Species	Strain	Species Designation	Taxonomy Classification		Genome Comparisons		
				MLST Analysis	Closest Relative	Nucleotide Identity (ANI)	DNA–DNA Hybridization	TYGS Type Strain
Cc-F	*Microbacterium xerophyticum*	AGC47	sp. nov	100%	*Microbacterium profundi*	89.1%	33%	No Match
Cc-R	*Microbacterium xerophyticum*	AGC85	sp. nov	100%	*Microbacterium profundi*	89.1%	33%	No Match
Cc-R	*Microbacterium umsixpiens*	AGC62	sp. nov	66%	*Microbacterium liquefaciens*	86.5%	44.4%	No Match

## Data Availability

Accession numbers for the 16S rRNA gene sequences are available in GenBank under PV706301 and PV706316. Genome sequence data for the bacterial isolates have been deposited in the Sequence Read Archive (SRA) under SRR29855773, SRR29855759, and SRR29855758. The type strains of the two novel species have been deposited in the Moroccan Coordinated Collections of Micro-organisms (CCMM) (https://www.ccmm.ma, accessed on 5 November 2025), with accession numbers B1344T (*Microbacterium xerophyticum* sp. nov.) and B1339T (*Microbacterium umsixpiens* sp. nov.). The genome assemblies are publicly accessible on Zenodo (https://zenodo.org/ accessed on 3 November 2025) under the following DOIs: https://doi.org/10.5281/zenodo.17503980.

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
