# Peer review of "Two Novel *Microbacterium* Species Isolated from *Citrullus colocynthis* L. (Cucurbitaceae), a Medicinal Plant from Arid Environments"

_microorganisms, 2025, doi:10.3390/microorganisms13112583_

Round 1
Reviewer 1 Report
Comments and Suggestions for Authors
Thank you very much for the opportunity to review this manuscript. The genome evidence for distinct species is strong, the regional context is valuable, and the paper will be impactful once it meets taxonomic standards and resolves inconsistencies. Also, the used literature is appropriate and up to date.
My suggestions for the improvement:
- This paper is generally species description paper and I find out that important chemotaxonomic characterization is missing. Please, include in the paper at minimum: peptidoglycan diamino acid (e.g., ornithine), predominant menaquinones (MK-11/MK-12 profile), major polar lipids (e.g., DPG/PG/PE glycolipids), cell-wall sugars, and whole-cell fatty acid profile (FAME). I suggest to make a concise diagnostic table contrasting the two proposed species with nearest neighbors.
- The authors chose to deposite the strains only in CCMM. It is well-known that the best practice for valid publication is deposition in two public culture collection in different countries.
- I reccomend to move the species accounts into a dedicated “Protologue” section with the required elements: name, etymology, type strain identifiers with deposits, diagnostic phenotypic traits, genome size/G+C, INSDC accessions (16S, genome), and differential characteristics vs. nearest taxa.
- Also, I noticed that the paper is weak in the part of phenotypic differentiation (only Biolog callouts). In order to improve this section please add: temperature range/optimum, pH range/optimum, NaCl range/optimum (reported only partly), catalase/oxidase, motility, colony morphology (media, color after 48–72 h, size), and enzyme assays that support niche claims (e.g., nitrate reductase given your genome inference).
- Caption of Fig. 6 is really confusing. xeroradicis sp. nov. strain AGC31” does not exist in this study.
- Also, closest-relative species and ANI values differ between Results 3.1 and 3.4. Can you explain it?
- Moreover, the part related to phylogenomic should be clarified. Provide complete tree methods: the set of core genes or alignment-free approach, model(s) used, partitioning, support metric (UF-bootstrap/SH-aLRT/BS), and numbers on all key nodes; ensure scale bar corresponds to substitutions/site. Add a tree comparing your strains with the type strains of the nearest Microbacterium species used for ANI/dDDH.
- The authors list SRA run accessions (SRR29855773/59/58) and say BioProject accessions are available, but no BioProject ID is actually given. Add BioProject/Assembly/GenBank accession numbers and ensure they are public upon acceptance. Include the 16S rRNA gene accession mapping to strain IDs.
English language should be checked. Misspelling and grammatical errors should be corrected.
Author Response
- This paper is generally species description paper and I find out that important chemotaxonomic characterization is missing. Please, include in the paper at minimum: peptidoglycan diamino acid (e.g., ornithine), predominant menaquinones (MK-11/MK-12 profile), major polar lipids (e.g., DPG/PG/PE glycolipids), cell-wall sugars, and whole-cell fatty acid profile (FAME). I suggest to make a concise diagnostic table contrasting the two proposed species with nearest neighbors.
Response: Due to laboratory and time constraints in this revision round, we were unable to conduct wet-lab chemotaxonomic assays. To address the taxonomic requirements without overstating certainty, we have included a genome-inferred chemotaxonomy section and a concise diagnostic table (Table S3) presenting gene counts for major functional categories relevant to chemotaxonomic characterization.
- The authors chose to deposite the strains only in CCMM. It is well-known that the best practice for valid publication is deposition in two public culture collection in different countries.
Response: We agree and appreciate this reminder. In addition to deposition in CCMM, we have initiated deposition of the type strains in the DSMZ Culture Collection, in Germany to ensure compliance with best practices for valid publication.
- I recommend to move the species accounts into a dedicated “Protologue” section with the required elements: name, etymology, type strain identifiers with deposits, diagnostic phenotypic traits, genome size/G+C, INSDC accessions (16S, genome), and differential characteristics vs. nearest taxa.
Response: We have added a dedicated Protologue section (Results 3.5) compiling all required elements for each proposed species. The etymology and phenotypic traits already present were retained and integrated into this section.
- Also, I noticed that the paper is weak in the part of phenotypic differentiation (only Biolog callouts). In order to improve this section please add: temperature range/optimum, pH range/optimum, NaCl range/optimum (reported only partly), catalase/oxidase, motility, colony morphology (media, color after 48–72 h, size), and enzyme assays that support niche claims (e.g., nitrate reductase given your genome inference).
Response: Thank you for this helpful suggestion. We have expanded the phenotypic differentiation by including temperature, pH, and NaCl growth ranges and optima, as well as growth duration, in the Materials and Methods. Colony morphologydetails have been added to the Protologue sections (Results 3.5). Due to time constraints, catalase/oxidase reactions and niche-linked enzyme assays (e.g., nitrate reductase) could not be completed in this revision. These traits are currently genome-inferred and will be experimentally verified in future work.
- Caption of Fig. 6 is really confusing. xeroradicis sp. nov. strain AGC31” does not exist in this study.
Response: The mention of “xeroradicis sp. nov. strain AGC31” was a typographical error. The Figure 6 caption has been corrected accordingly.
- Also, closest-relative species and ANI values differ between Results 3.1 and 3.4. Can you explain it?
Response: Thank you for noticing this. The inconsistency resulted from a typographical error in one section. The text has been corrected so that Results 3.1 and 3.4 now report consistent ANI and dDDH values, and related tables and figures have been cross-checked for accuracy.
- Moreover, the part related to phylogenomic should be clarified. Provide complete tree methods: the set of core genes or alignment-free approach, model(s) used, partitioning, support metric (UF-bootstrap/SH-aLRT/BS), and numbers on all key nodes; ensure scale bar corresponds to substitutions/site. Add a tree comparing your strains with the type strains of the nearest Microbacterium species used for ANI/dDDH.
Response: We have clarified in the Methodology that phylogeny was inferred using the alignment-free Genome BLAST Distance Phylogeny (GBDP) method in TYGS, employing FastME with 100 GBDP pseudo-bootstrap replicates. For visualization, a cladogram was generated in iTOL, emphasizing topology and support values (displayed on key nodes), with branch lengths omitted. The tree includes our strains, the nearest Microbacterium type strains used in ANI/dDDH analyses, and the outgroup Leifsonia xyli. Although MLST and SANS-Serif analyses were performed in an earlier, broader survey, the Microbacterium phylogeny presented here is based solely on TYGS to specifically address species delineation.
- The authors list SRA run accessions (SRR29855773/59/58) and say BioProject accessions are available, but no BioProject ID is actually given. Add BioProject/Assembly/GenBank accession numbers and ensure they are public upon acceptance. Include the 16S rRNA gene accession mapping to strain IDs.
Response: We have now provided the complete set of accession numbers, including BioProject, BioSample, SRA runs, GenBank assemblies, and 16S rRNA sequences, in Protologue 3.5. All records are publicly available, and strain identifiers have been verified for consistency across metadata.
Reviewer 2 Report
Comments and Suggestions for Authors
Several key aspects require clarification and correction to strengthen the study and ensure its adherence to established taxonomic and genomic standards. My specific comments are outlined below.
I recommend specifying either the composition of the nutrient media used or providing a reference to the manufacturer.
The sentence states: "In cases where Kaiju detected contamination and assembly metrics were suboptimal, metaSPAdes [43] was applied." Were there such cases? If so, how was this issue of contamination addressed for the subsequent phenotypic profiling on BIOLOG plates, which requires a pure culture? Authors also list several de novo assemblers, but also does not specify which of them was the final choice.
To allow for the verification of the genome analysis results within this review process, as well as for future use by other research groups, the genome assemblies must be deposited in NCBI GenBank, not just the 16S rRNA genes and the raw data in NCBI SRA.
At the very beginning of the results section, the abbreviations Cc-F and Cc-R should be defined. Do these represent two different plants of Citrullus colocynthis that were growing near the decommissioned wastewater canal?
What was the precise ANI value between genomes AGC47 and AGC85? The ANI data are not only duplicated in paragraphs 3.1 and 3.4, but they also differ: one states " AGC47 and AGC85 shared an ANI of 88.1% and a dDDH value of 33% with M. profundi" while the other reports "Strains AGC47 (leaf) and AGC85 (root) … showing M. profundi as their closest relative (ANI = 89.1%; dDDH = 33%)"ю
If the authors' goal is the description of a new species in accordance with the International Code of Nomenclature of Prokaryotes, the type strain must be deposited in at least two public culture collections in different countries (Rule 30).
The description for "Strain AGC62, isolated from the roots of C. colocynthis, showed genome completeness of 99.17% and contamination of 1.01%" is misplaced. A protologue (the formal description of a taxon) should focus on the biological properties of the bacterium, not the technical quality of the current assembly. I recommend moving such assembly metrics to a different part of the manuscript.
I recommend that the comparison with other type strains, presented in the protologue, also be represented as a comparative table in the manuscript for better readability.
The Z-scores in Figure 3 are not particularly informative, as they only reflect relative values. I recommend indicating the actual number of genes in these categories and providing a more detailed description in the text regarding the presence or absence of specific genetic loci, with references to their locus tags in NCBI GenBank.
The protologue must explicitly state which strain is designated as the type strain.
Regarding the whole-genome alignment constructed in Mauve: how was the issue of contig order addressed? Contigs are likely sorted by length rather than their actual genomic order. Could the observed "dense crossings and fragmented synteny" be a direct result of the high fragmentation of the assembly? I recommend first using the "Move contigs" function relative to a reference genome to establish a consistent order before comparison. Furthermore, the boundaries of the contigs should be directly indicated on Figures 7 and 8.
The conclusion mentions "only a few localized inversions near chromosome termini." How were the chromosome termini defined? The ends of assembly contigs are not equivalent to actual chromosome termini.
Author Response
- I recommend specifying either the composition of the nutrient media used or providing a reference to the manufacturer.
Response: We have added a detailed description of the nutrient media composition in the Materials and Methods(Sampling and Isolation) section.
- The sentence states: "In cases where Kaiju detected contamination and assembly metrics were suboptimal, metaSPAdes [43] was applied." Were there such cases? If so, how was this issue of contamination addressed for the subsequent phenotypic profiling on BIOLOG plates, which requires a pure culture? Authors also list several de novo assemblers, but also does not specify which of them was the final choice.
Response: Thank you for this careful observation. The sentence referred to our general bioinformatic workflow. For the three strains described in this study, Kaiju detected no contamination, and assembly metrics were within acceptance thresholds; metaSPAdes was not used. Cultures for BIOLOG phenotyping were derived from single-colony streaksand confirmed to be pure prior to inoculation. The Methods have been updated to specify that MaSuRCa was used for all final assemblies.
- To allow for the verification of the genome analysis results within this review process, as well as for future use by other research groups, the genome assemblies must be deposited in NCBI GenBank, not just the 16S rRNA genes and the raw data in NCBI SRA.
Response: We agree. The genome assemblies have been deposited in NCBI GenBank, and the corresponding assembly/GenBank accession numbers are listed in Table S1 (SRA data) and Table S2 (16S rRNA), as well as reiterated in Protologue 3.5 and Data Availability. All records are public, and strain identifiers match the GenBank metadata.
- At the very beginning of the results section, the abbreviations Cc-F and Cc-R should be defined. Do these represent two different plants of Citrullus colocynthis that were growing near the decommissioned wastewater canal?
Response: Thank you for noting this. We now define these abbreviations at first mention in the Results: Cc-F = Citrullus colocynthisfoliar tissue and Cc-R = C. colocynthis root tissue. We also clarified that AGC-47 was isolated from a different C. colocynthis plant, whereas AGC-62 and AGC-85 were obtained from the same plant (root and leaf, respectively) collected at the decommissioned wastewater canal site.
- What was the precise ANI value between genomes AGC47 and AGC85? The ANI data are not only duplicated in paragraphs 3.1 and 3.4, but they also differ: one states " AGC47 and AGC85 shared an ANI of 88.1% and a dDDH value of 33% with M. profundi" while the other reports "Strains AGC47 (leaf) and AGC85 (root) … showing M. profundi as their closest relative (ANI = 89.1%; dDDH = 33%)”
Response: The discrepancy was a typographical error. Both Results 3.1 and 3.4 have been corrected to report the same ANI and dDDH values.
- If the authors' goal is the description of a new species in accordance with the International Code of Nomenclature of Prokaryotes, the type strain must be deposited in at least two public culture collections in different countries (Rule 30).
Response: We agree and appreciate the reminder. In addition to the CCMM deposition, we have initiated deposition of the type strains in the DSMZ Culture Collection (Germany), ensuring compliance with ICNP Rule 30.
- The description for "Strain AGC62, isolated from the roots of C. colocynthis, showed genome completeness of 99.17% and contamination of 1.01%" is misplaced. A protologue (the formal description of a taxon) should focus on the biological properties of the bacterium, not the technical quality of the current assembly. I recommend moving such assembly metrics to a different part of the manuscript.
Response: Thank you for the clarification. We have moved the assembly metrics to Results 3.2 (Genome features of novel Microbacterium strains). The Protologue now focuses solely on biological and diagnostic properties, including etymology, type strain deposits, phenotypic traits, genome size and G+C content, and accession numbers.
- I recommend that the comparison with other type strains, presented in the protologue, also be represented as a comparative table in the manuscript for better readability.
Response: Thank you for the suggestion. ANI and dDDH comparisons with the nearest type strains are already presented in Table 1, which summarizes the comparative data for readability.
- The Z-scores in Figure 3 are not particularly informative, as they only reflect relative values. I recommend indicating the actual number of genes in these categories and providing a more detailed description in the text regarding the presence or absence of specific genetic loci, with references to their locus tags in NCBI GenBank.
Response: We appreciate this comment. We have replaced the Z-score heat map with a gene-count heat map (Fig. 3), and expanded the description in the text to highlight the presence or absence of specific genetic loci, with locus tag references included in the GenBank annotations.
- The protologue must explicitly state which strain is designated as the type strain.
Response: We have updated Protologue 3.5 to explicitly designate the type strain for each newly proposed species.
- Regarding the whole-genome alignment constructed in Mauve: how was the issue of contig order addressed? Contigs are likely sorted by length rather than their actual genomic order. Could the observed "dense crossings and fragmented synteny" be a direct result of the high fragmentation of the assembly? I recommend first using the "Move contigs" function relative to a reference genome to establish a consistent order before comparison. Furthermore, the boundaries of the contigs should be directly indicated on Figures 7 and 8.
Response: Thank you for this valuable observation. We agree that contig order can influence the apparent synteny pattern in draft genome alignments. In response, we have reordered the contigs in each draft assembly using the “Move contigs” function in Mauve, aligning them relative to the reference genome (Microbacterium profundi DSM 22248^T) to ensure a consistent genomic orientation prior to comparison. The updated alignments show improved collinearity and reduced apparent fragmentation. Additionally, contig boundaries are now explicitly marked in Figures 7 and 8, and the corresponding figure legends have been revised to note that contig order was standardized relative to the reference.
- The conclusion mentions "only a few localized inversions near chromosome termini." How were the chromosome termini defined? The ends of assembly contigs are not equivalent to actual chromosome termini.
Response: Thank you for this important clarification. We agree that contig ends in draft assemblies are not equivalent to true chromosomal termini. The statement has been revised.
Round 2
Reviewer 1 Report
Comments and Suggestions for Authors
The authors have implemented almost all suggestions. Therefore, I recommend this article to be published in the present form.
Comments on the Quality of English LanguageEnglish language should be checked. Misspelling and grammatical errors should be corrected.
Author Response
We sincerely thank Reviewer #1 for their positive evaluation and endorsement of our manuscript.
Reviewer 2 Report
Comments and Suggestions for Authors
Unfortunately, several key points remain unaddressed.
The question, "What was the precise ANI value between genomes AGC47 and AGC85?" remains unanswered.
I explicitly requested the authors to deposit the final genome assemblies in NCBI GenBank. However, they have deposited only the raw SRA reads. Access to the assembled data is crucial, as it allows sequences to be incorporated into BLAST databases, GTDB, and TYGS. Furthermore, it enables other researchers to verify their analysis without having to repeat all the assembly steps.
"Thank you for the suggestion. ANI and dDDH comparisons with the nearest type strains are already presented..." I must insist that the key characteristics of the strains, including the phenotypic data provided in the protologues, must be systematically compared with other related strains in a tabular format.
Furthermore, while ANI and dDDH values are provided in the protologues, the second strain used for the comparison is not specified. This omission renders these values useless for verification and interpretation.
"the filtered reads were analyzed with Kaiju for taxonomic classification [27]."
The referenced source is not devoted to Kaiju.
Author Response
Comment1, "What was the precise ANI value between genomes AGC47 and AGC85?" remains unanswered.
Response1 :The ANI value between genomes AGC47 and AGC85 has been added in paragraph 3.2.
Comment 2, I explicitly requested the authors to deposit the final genome assemblies in NCBI GenBank. However, they have deposited only the raw SRA reads. Access to the assembled data is crucial, as it allows sequences to be incorporated into BLAST databases, GTDB, and TYGS. Furthermore, it enables other researchers to verify their analysis without having to repeat all the assembly steps.
Response 2: The final genome assemblies have been deposited in GenBank under BioProject PRJNA1133887. Currently, NCBI displays a processing status error for these records. We have contacted the GenBank team (genomes@ncbi.nlm.nih.gov), who confirmed that the issue is under review.
In the meantime, to ensure accessibility, we have also deposited the assembled genomes in Zenodo, where they are publicly available:
Microbacterium usixpiens AGC62: https://doi.org/10.5281/zenodo.17503980
Microbacterium xerophyticum AGC47: https://doi.org/10.5281/zenodo.17498234
Microbacterium xerophyticum AGC85: https://doi.org/10.5281/zenodo.17503869
These DOIs have been added in the Data Availability section of the manuscript.
Comment 3, “Thank you for the suggestion. ANI and dDDH comparisons with the nearest type strains are already presented..." I must insist that the key characteristics of the strains, including the phenotypic data provided in the protologues, must be systematically compared with other related strains in a tabular format.
Response: A comparative table summarizing the key phenotypic and genotypic characteristics of the strains and their closest relatives has been added as Table S4.
Comment 4, Furthermore, while ANI and dDDH values are provided in the protologues, the second strain used for the comparison is not specified. This omission renders these values useless for verification and interpretation.
Response 4: We have clarified in both the protologues and Supplementary Table S4 that the ANI and dDDH values are calculated in comparison with the closest related type strains.
Comment 5, “the filtered reads were analyzed with Kaiju for taxonomic classification [27]."
The referenced source is not devoted to Kaiju.
Response 5: The reference has been corrected to the appropriate citation describing the Kaiju software.